# Concordance between whole exome sequencing of circulating tumor DNA and tumor tissue

**Julanee Leenanitikul**[1], **Prangwalai Chanchaem**[2], **Suwanan Mankhong**[2], **Sikrit Denariyakoon**[3], **Valla Fongchaiya**[3], **Areeya Arayataweegool**[3], **Pattama Angspatt**[4], **Ploytuangporn Wongchanapai**[4], **Verayuth Prapanpoj**[5], **Kris Chatamra**[3], **Trairak Pisitkun**[6,7], **Sira Sriswasdi**[7,8]*, **Piriya Wongkongkathep**[6,7]*

1 Bioinformatics and Computational Biology Program, Chulalongkorn University, Bangkok, Thailand, 2 Research Unit of Systems Microbiology, Faculty of Medicine, Chulalongkorn University, Bangkok, Thailand, 3 The Queen Sirikit Center for Breast Cancer, King Chulalongkorn Memorial Hospital, Thai Red Cross Society, Bangkok, Thailand, 4 Division of Medical Oncology, Department of Medicine, Faculty of Medicine, Chulalongkorn University and the King Chulalongkorn Memorial Hospital, Bangkok, Thailand, 5 Medical Genetics Center, Bangkok, Thailand, 6 Center of Excellence in Systems Biology, Faculty of Medicine, Chulalongkorn University, Bangkok, Thailand, 7 Research Affairs, Faculty of Medicine, Chulalongkorn University, Bangkok, Thailand, 8 Center of Excellence in Computational Molecular Biology, Faculty of Medicine, Chulalongkorn University, Bangkok, Thailand

* sira.sr@chula.ac.th (SS); piriya.w@chula.ac.th (PW)

**Data Availability Statement:** The full mutational profiles and metadata for all patients have been provided as supplementary table (in excel format). We do not share raw exome sequencing data because they are considered personal and private

## Abstract

Next generation sequencing of circulating tumor DNA (ctDNA) has been used as a noninvasive alternative for cancer diagnosis and characterization of tumor mutational landscape. However, low ctDNA fraction and other factors can limit the ability of ctDNA analysis to capture tumor-specific and actionable variants. In this study, whole-exome sequencings (WES) were performed on paired ctDNA and tumor biopsy in 15 cancer patients to assess the extent of concordance between mutational profiles derived from the two source materials. We found that up to 16.4% ctDNA fraction can still be insufficient for detecting tumor-specific variants and that good concordance with tumor biopsy is consistently achieved at higher ctDNA fractions. Most importantly, ctDNA analysis can consistently capture tumor heterogeneity and detect key cancer-related genes even in a patient with both primary and metastatic tumors.

## Introduction

Next-generation sequencing (NGS) has been frequently applied to detect clinically actionable somatic mutations from tumor tissues [1, 2]. Tissue biopsy is a standard method for acquiring tumor tissues to characterize tumor molecular profile. However, tissue biopsy is invasive and, in some cases, cannot be performed due to poor patient's health. Additionally, there are limitations in terms of tissue accessibility, the impracticality of performing multiple biopsies over time, and the inability of a single biopsy to capture intra-tumor heterogeneity [3–7]. Therefore, non-invasive liquid biopsy, such as analysis of tumor DNA in blood sample, has become a promising alternative for cancer precision medicine.

to the patients. The following Data Availibity statement has been added to the manuscript and provided with a new cover letter: The minimal data set required to reanalyze or reproduce the findings of this study has been provided in supplementary materials. Sample information, including cancer types, ctDNA fractions, and sampling time points, are provided in S1 Table in S1 File. The full lists of filtered variants identified in each sample together with their allele frequencies are organized by cancer types and provided in S4, S5, and S6 Tables S1 File. The full lists of variants of top cancer-related genes are organized by cancer types and provided in S8, S9, and S10 Tables S1 File. Raw exome sequencing data are considered patient's private information but may be made available upon a reasonable request to the corresponding author.

**Funding:** This research was partially supported by the Thailand Science Research and Innovation Fund, Chulalongkorn University (HEA663000041 to S.S.), the grants for development of new faculty staff, Ratchadaphiseksomphot Fund, Chulalongkorn University (to P.W.), and Ratchadaphiseksomphot Fund, Faculty of Medicine, Chulalongkorn University, grant number RA65/029 (to P.W., S.S., and T.P.). The funders had no role in study design, data collection and analysis, decision to publish, or preparation of the manuscript.

**Competing interests:** The authors declare no competing interests.

Liquid biopsy of blood samples generally detects circulating tumor DNA (ctDNA) or circulating tumor cells (CTC). NGS of ctDNA has gained popularity in clinical practice in recent years. ctDNA mainly originates from apoptosis, necrosis of dead cancer cells, or secretion of living cancer cells into the bloodstream [8–10]. Hence, tumor profiling with ctDNA has the potential to not only reflect tumor heterogeneity but also capture information from both primary tumor and distant metastasis that would be missed by a single biopsy. Furthermore, since obtaining ctDNA is non-invasive, liquid biopsies can be performed multiple times to monitor cancer progression and response to treatment. However, clinical utilization of ctDNA is still limited by a lack of standardized technique and insufficient validation [11–13]. Because ctDNA fraction in plasma can vary from 0.01% to 90% in total cell-free DNA, whole exome sequencing (WES) of ctDNA can be inapplicable in many patients. Instead, targeted sequencing of a small set of actionable genes, such as EGFR, is often performed on ctDNA in clinical settings [14, 15]. As a result, several analysis parameters and tumor characteristics [16] that can influence WES of ctDNA has not been thoroughly investigated. These issues need to be considered for developing a new test.

To screen patients for which WES of ctDNA may be applicable, ultra-low pass whole genome sequencing (ULP-WGS) can be performed first evaluate the ctDNA fraction of a patient. WES of ctDNA would then be performed only if the ctDNA fraction reaches a certain cutoff. A prior study has shown that applying this criterion yielded ctDNA WES data that is concordant with traditional WES of tumor biopsy and capable of detecting metastatic tumors [17]. Here, we performed comparative analysis of WES of ctDNA and tumor biopsy from patients with various cancer types, a broad range of ctDNA fractions, and different time interval between tissue biopsy and blood draw to evaluate the impact of these factors on the degree of concordance between mutational profiles derived from WES of the two source materials. Our study also showed that WES ctDNA can capture the mutational profiles of both primary and metastatic tumor sites within the same patient.

## Materials and methods

### Patient sample collection

Patients were enrolled in a Breast Cancer cohort at the Queen Sirikit Center for Breast Cancer (QSCBC) and a multi-tumor cohort at the Medical Oncology Unit (MedOnco), King Chulalongkorn Memorial Hospital, Thai Red Cross Society in Bangkok, Thailand. This project was approved by the Institutional Review Board Committee, Faculty of Medicine, Chulalongkorn University (IRB No. 068/60). Patients in the QSCBC cohort were recruited between July 2019 and June 2021 and patients in the MedOnco cohort were recruited between September 2020 and February 2021. Subsequent ctDNA, tumor DNA, and normal DNA samples were collected between July 2019 and November 2021. Full collection time points are provided in S1 Table in S1 File. Blood samples for ctDNA WES of patients were collected within 4 months after blood sample collection for ctDNA ULP-WGS (S1 Table in S1 File, median = 2 months). The length of time interval between tumor tissue and ctDNA collections was within 5 months (median = 2 months). Written informed consents were obtained from all participants.

### Sample processing and DNA extraction

Solid tumor tissues obtained from tumor biopsy or surgery were collected by snap-frozen technique with liquid nitrogen and homogenized using QIAGEN TissueLyser. Normal DNA samples were obtained from peripheral blood mononuclear cells (PBMC) collected in an ethylene diamine tetra-acetic acid (EDTA) tube. Blood samples were first centrifuged at 1600x g for 15 minutes at 4˚ C to separate buffy coats. DNA from tumor and normal cells were then extracted

using QIAGEN AllPrep kit. To collect ctDNA, the blood samples were collected in a Streck tube and centrifuged at 1600x g for 15 minutes at 4° C to isolate plasma. MagMax ctDNA isolation kit was used for ctDNA extraction. The amounts of DNA were quantified using Qubit fluorometer.

## Ultra-low pass whole genome sequencing

To assess the tumor DNA fraction in ctDNA sample, ctDNA were size-selected using AMPure XP Beads. The libraries were prepared using Ion Plus Fragment Library Kit and sequenced by an Ion OneTouch 2 system coupled to a single-ended Ion Proton Sequencer. Reads from ULP-WGS of ctDNA samples were mapped to the hg 19 (GRCh37) reference genome to calculate genome-wide coverages. Tumor DNA fractions were calculated using the ichorCNA software [17].

## Whole exome sequencing

The libraries were prepared using Agilent SureSelectXT Human All Exon V6 and COSMIC kit or SureSelect Human All Exon V7 kit for whole-exome sequencing and sequenced using Illumina HiSeq or MGI MGISE-200 platform at 2 x 150bp pair-ended configuration. Sequenced data were aligned to the hg38 (GRCh38) reference genome using the Burrows-Wheeler Aligner (BWA) [18] and processed using the Genome Analysis Toolkit (GATK) [19]. Germline variants in PBMC samples were called with HaplotypeCaller [20]. Somatic variants in ctDNA and tumor biopsy samples were called using 3 tools, Mutect2 [21], Strelka [22], and VarScan [23]. Variant Call Format (VCF) files from 3 different variant callers were merged into a single VCF file for each patient and subjected to quality filtering. A called variant was removed if (1) the variant allele frequencies (VAF) in both ctDNA and tumor samples were lower than 1%, (2) the variant allele frequency in the normal sample was higher than 5%, (3) there were less than 4 reads supporting the variant, or (4) the sequencing depth was less than 8 reads. Cancer relevant genes and variants were defined according to the Catalogue of Somatic Mutation in Cancer (COSMIC) database [24, 25].

## Concordance analysis of mutational profiles

The concordance between mutational profiles derived from WES of ctDNA and tumor tissue was calculated at two levels. The first level is based on the Jaccard similarity index of the number of shared variants (i.e., identified in both sample sources) that passed the quality filtering defined above. This measures whether the two WES analyses identified the same set of variants. Variants identified exclusively in either sample were considered discordant. The second level is based on the Pearson's correlation coefficient of variant allele frequencies (VAF) of shared variants. This measures whether the two WES analyses define variants as abundant or rare in the same manner. For the melanoma patient with a tumor metastatic site (MN01), the concordance between ctDNA and each tumor was evaluated separately.

## Subclones identification and analysis

Clustering of identified variants in each WES result into subclones was performed using MAGOS [26]. Subclones containing fewer than 3 unique variants were removed. To map similar subclones identified in WES of tumor tissue and ctDNA, Jaccard Index were calculated pairwise between each set of variants associated with a subclone in tumor tissue and each set of variants associated with a subclone in ctDNA. For each ctDNA subclone, the corresponding

subclone in tumor tissue with the highest Jaccard Index was defined putatively as the same subclone.

## Statistical analysis

Standard descriptive statistics were used to characterize the level of overlap between ctDNA and tumor, the mutation landscape, and the profile of base substitutions (e.g., C>T/G>A, T>G/A>C) identified in each patient and cancer type. Top genes in mutation landscape study were selected by tissue type from the Cancer Gene Census on COSMIC [27]. The mutation landscape was visualized using Maftools [28] (version 2.8.0) in R (version 4.1.3). Standard inferential statistics and Pearson's correlation coefficients were considered significant if the p-values were less than 0.01. Venn diagrams were visualized using ggvenn (version 0.1.9), and ggplot2 (version 3.3.6) in R.

# Results

## Patient characteristics

ctDNA and tumor whole exome sequencing (WES) data from 15 cancer patients with 4 cancer types were analyzed (Table 1). There are nine patients with breast cancer, one patient with sarcoma, four patients with gastrointestinal cancer (one cecum cancer, two colon cancer, and one duodenum cancer), and one melanoma patient with tissues collected from both the primary skin site and a metastatic liver site. The time interval between ctDNA and tumor tissue sample collections varied from 0 to 17 months, with a median of 3 months, due to external logistical factors (S1 Table in S1 File and S1 Fig). The data collection workflow starts with an ultra-low pass whole-genome sequencing (ULP-WGS) of the blood samples to assess ctDNA fractions. WES was performed on ctDNA from patients with more than 7% ctDNA fractions (Fig 1). It should be noted that the choice of 7% ctDNA fraction cutoff should not affect the findings because none of the analyzed samples with up to 16.4% ctDNA fractions yield enough tumor-specific variants (Fig 2A).

## Impact of ctDNA tumor fraction on exome sequencing

To quantify the impact of ctDNA fraction on the quality of subsequent exome sequencing results, relationships between ctDNA fractions estimated from ULP-WGS and the numbers and allele frequencies of non-synonymous variants identified from WES of ctDNA and tumor tissue were analyzed. There is a positive trend between the number of identified variants and ctDNA fraction, but the correlation is not significant, likely due to the high variation and low number of identified variants among samples with ctDNA fractions above 50% (Fig 2A, Pearson's correlation = 0.4657, p-value = 0.069). Very low numbers of non-synonymous variants

**Table 1. Patient characteristics.**

| Cancer type | Number of samples |
|---|---|
| **Breast** | 9 (60%) |
| **Sarcoma** | 1 (6.67%) |
| **Gastrointestinal** | 4 (26.67%) |
| • **Cecum** | 1 |
| • **Colon** | 2 |
| • **Duodenum** | 1 |
| **Melanoma** | 1 (6.67%) |

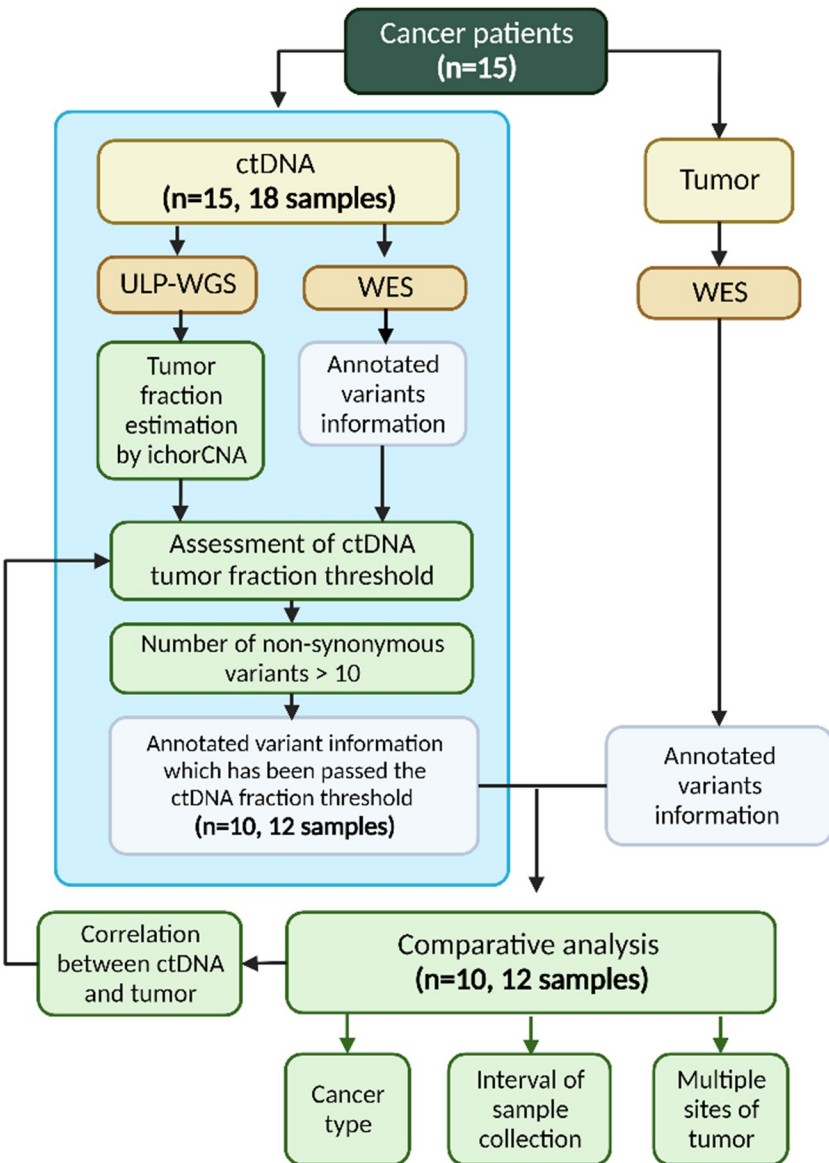

**Fig 1. The study workflow of 15 cancer patients.** Blue box indicates ctDNA protocol and analysis. After a preliminary round of WES quality check, 6 samples with low numbers of identified non-synonymous variants were also excluded from comparative analysis with tumor WES.

(≤10) were identified in all five samples with ≤16.4% ctDNA fractions (four breast cancer samples and one sarcoma sample). None of the gastrointestinal cancer samples exhibited poor ctDNA fraction. When the ctDNA fractions are high enough (>30% here), the number of identified variants no longer scales with ctDNA fraction. The median number of identified non-synonymous variants for samples with ≤16.4% ctDNA fractions is 3 variants while the

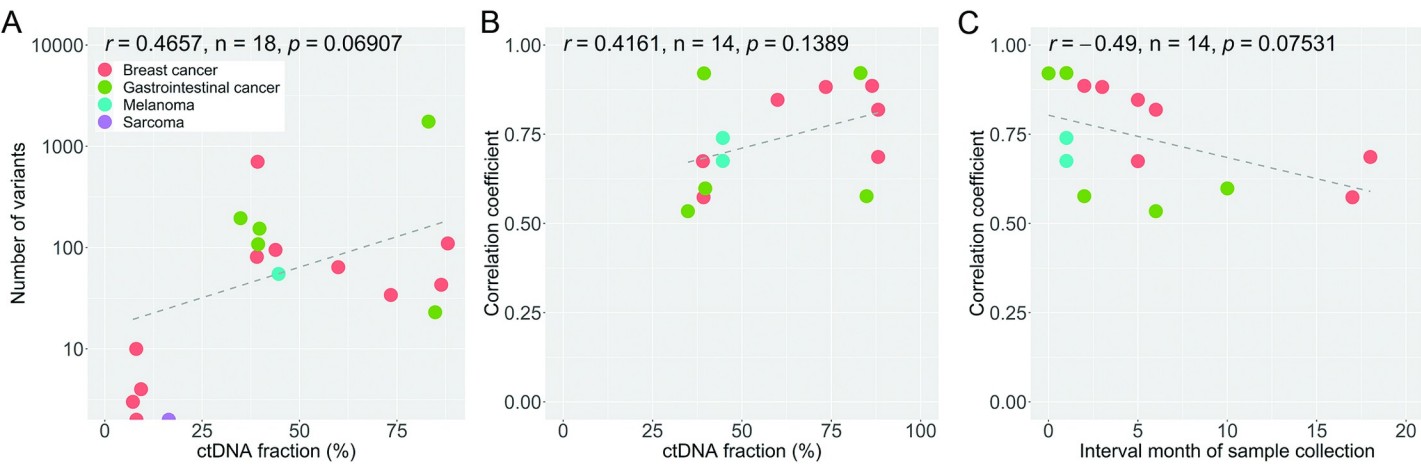

**Fig 2. Assessment of ctDNA tumor fraction threshold.** Each dot represents a sample or a pair of samples (BC: breast cancer, GC: gastrointestinal cancer, SC: sarcoma, MN: melanoma). The numbers following *tm* (tumor) and *ct* (ctDNA) indicate the sample collection time points, which are provided in S1–S3 Tables in S1 File. The overall Pearson's correlation coefficients (r), sample sizes (n), and p-values are indicated. A) Comparison between estimated ctDNA fraction and the number of non-synonymous variants identified in ctDNA WES. B) Comparison between ctDNA fraction and the degree of concordance (Pearson's correlation coefficient) between variant allele frequency estimated from ctDNA and from tumor tissue. Data from samples with at least 20 non-synonymous variants are shown. C) Comparison between the length of time interval between tumor tissue biopsy and ctDNA sample collection and the degree of concordance of estimated VAFs.

median for samples with >30% ctDNA fractions is 95 variants (S2 Table in S1 File). The chromosomal copy number profiles for the four patients with low ctDNA fractions were shown in S2 Fig. It should be noted that relaxing the variant filtering criteria only increases the numbers of called variants by 1–2 in these samples.

When focusing the samples with sufficient numbers (>20) of identified non-synonymous variants, there is also a positive, but non-significant, correlation between ctDNA fractions and the concordance between WES data from ctDNA and tumor tissue in term of allele frequencies (Fig 2B, Pearson's correlation = 0.42). In contrast, there is no correlation between ctDNA fraction and the Jaccard Index of the overlap in identified non-synonymous variants (S3 Table in S1 File, Pearson's correlation = 0.0067). For example, the ctDNA fraction of BC09 sample was only 39% but the WES data identified similar variants (Jaccard Index = 0.70, 61 shared and 26 discordant variants). On the other hand, the ctDNA fraction of GC02 sample was 85% but the WES data identified almost completely different variants (Jaccard Index = 0.15, 9 shared and 52 discordant variants). This discrepancy can occur because low-VAF variants in tumor tissue may be missed by WES analysis of ctDNA and because the profile of tumor DNA that were shed into the bloodstream can considerably differ from the profiles of DNA in the biopsied site. Indeed, the VAFs of discordant variants are significantly lower than the VAFs of concordant variants (S3 Fig, Mann-Whitney U test p-values < 2.14e-5). Additionally, tumor heterogeneity may introduce a strong bias when tumor biopsies are performed, whereas the ctDNA can represent multiple clonalities.

## The drift in mutational profiles over time

Another factor that affects the degree of concordance between WES results of ctDNA and tumor tissue is the length of time interval between sample collections, as mutational profile of the tumor can shift over time. Here, the average and median time intervals between sample collection are 4 and 3 months, respectively. The length of time interval between sample collection clearly negatively correlates with the concordance between WES data from ctDNA and tumor tissue in terms of both VAFs and the Jaccard Index of the overlap in identified non-

synonymous variants (Fig 2C, Pearson's correlation = −0.49 and −0.25, respectively). However, unlike ctDNA fraction, there is no clear cutoff for the length of time interval below which a high degree of concordance between WES results would be guaranteed. ctDNA and tumor tissue collected within 2 months can be as discordant as those collected 6 months apart with respect to both Jaccard Index and correlation in VAF (S3 Table in S1 File).

## Concordance between ctDNA WES and tumor WES

To comprehensively evaluate the degree of concordance between WES data from ctDNA and tumor tissue, data for breast cancer and gastrointestinal patients were analyzed separately because VAFs from the two cancer types were significantly different (S3 Fig, median tumor VAFs = 11% for breast cancer and 36% for gastrointestinal cancer). For patients with multiple sample collection time points, concordances were calculated between WES of ctDNA and tumor samples that were collected closest to each other to minimize the impact from the drift in mutational profile over time observed earlier (Fig 2C). This results in six ctDNA-tumor pairs from five breast cancer patients, four ctDNA-tumor pairs from four gastrointestinal cancer patients, and two ctDNA-tumor pairs from one melanoma patient (S3 Table in S1 File).

For breast cancer patients, the average Jaccard index for the overlap of identified non-synonymous variants from ctDNA WES and tumor WES is 0.59 (SD = 0.16), with average time interval between sample collection of 6 months. VAFs of shared variants estimated from ctDNA and tumor WES are moderately concordant with an overall Pearson's correlation of 0.78 (Fig 3A, n = 528 shared variants). The detailed lists of variants and VAFs identified in these patients are provided in S4 Table in S1 File. For gastrointestinal cancer patients, the average Jaccard index is slightly higher at 0.64 (SD = 0.34), which may partly be due to the shorter average time interval between sample collection of 2 months. The high standard deviation in Jaccard Index is due to an outlier (GC02). The concordant in VAF of shared variants are moderately high with an overall Pearson's correlation of 0.74 (Fig 3B, n = 1965 shared variants). The detailed lists of variants and VAFs identified in these patients are provided in S5 Table in S1 File.

## ctDNA captures mutational profiles of multiple tumor sites

For the melanoma patient with two tumor sites, a primary skin site and a metastatic liver site, tissues from both sites were collected within the same week, and ctDNA sample was collected in the following month. As expected of a mixed mutational profile, VAFs estimated from ctDNA only moderately correlate with VAFs estimated from the skin or the liver site (Fig 3C and S3 Table in S1 File, Pearson's correlation = 0.68–0.74, Jaccard Index = 0.23–0.47). Based on the overlap in identified non-synonymous variants, WES of ctDNA is slightly more concordant with WES of the liver site (32 shared variants and 35 discordant variants) than with the skin site (24 shared variants and 80 discordant variants). Furthermore, WES of ctDNA can capture all 22 variants that are shared between the two tumor sites as well as 12 additional variants that are unique to one tumor site (Fig 3D). The detailed lists of variants and VAFs identified in this patient are provided in S6 and S7 Tables in S1 File.

## WES of ctDNA and tumor tissue identify similar mutational landscapes

The ability of ctDNA WES to capture clinically-relevant characteristics of the tumor mutational landscape was evaluated based on whether ctDNA WES can identify similar base substitution frequencies, which could be matched against known mutational signatures [29] to identify the underlying molecular mechanism of cancer, and the same variants of the top

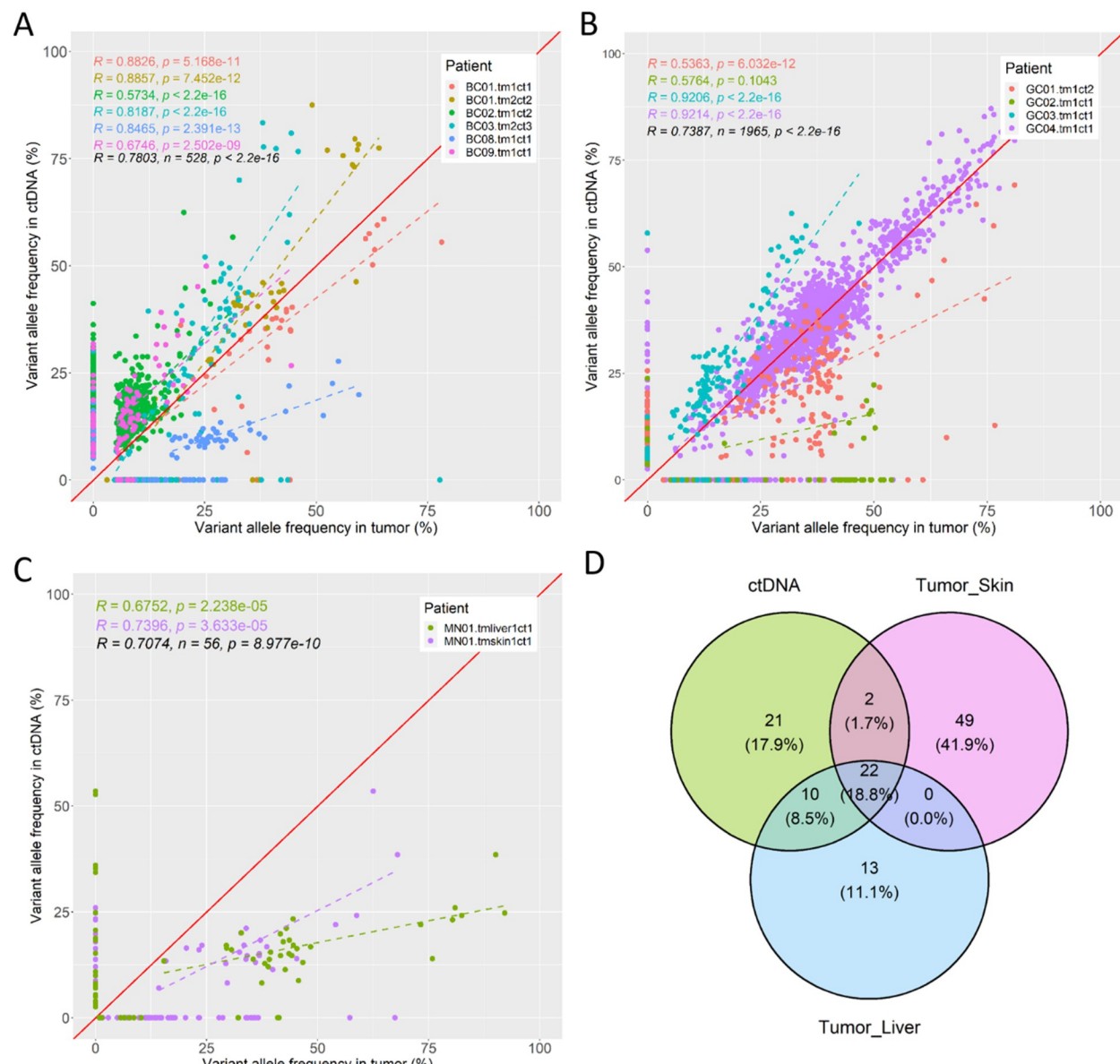

**Fig 3. Concordance between WES of ctDNA and tumor tissue.** Pearson's correlation coefficients and p-values are indicated. A) VAFs of non-synonymous variants identified in WES of ctDNA and tumor for each breast cancer patient. Dashed lines indicate the best linear fits. Each color represents a patient. B) Similar plot for gastrointestinal cancer patients. C) Similar plot for the melanoma patient with two tumor sites. D) Venn diagram of the overlap in non-synonymous variants identified from ctDNA and the two tumor sites of the melanoma patient.

cancer-related genes as tumor WES. Top genes for each cancer type were downloaded from the Cancer Gene Census [27] via COSMIC database (also called tier-1 genes).

In almost all cases, the same variants of tier-1 genes (Fig 4, same heatmap color between ctDNA and tumor tissue of the same patient) and the same base substitution profiles (Fig 4, bar plot at the bottom of each panel) identified in tumor WES could also be identified in ctDNA WES. For breast cancer patients, known oncogenes and tumor suppressor genes such as ESR1, KRAS, PIK3CA, PIK3R1, FAT1, and MED12 were consistently identified in both ctDNA and tumor WES across patients (Fig 4A). The few discordant cases, such as the missing

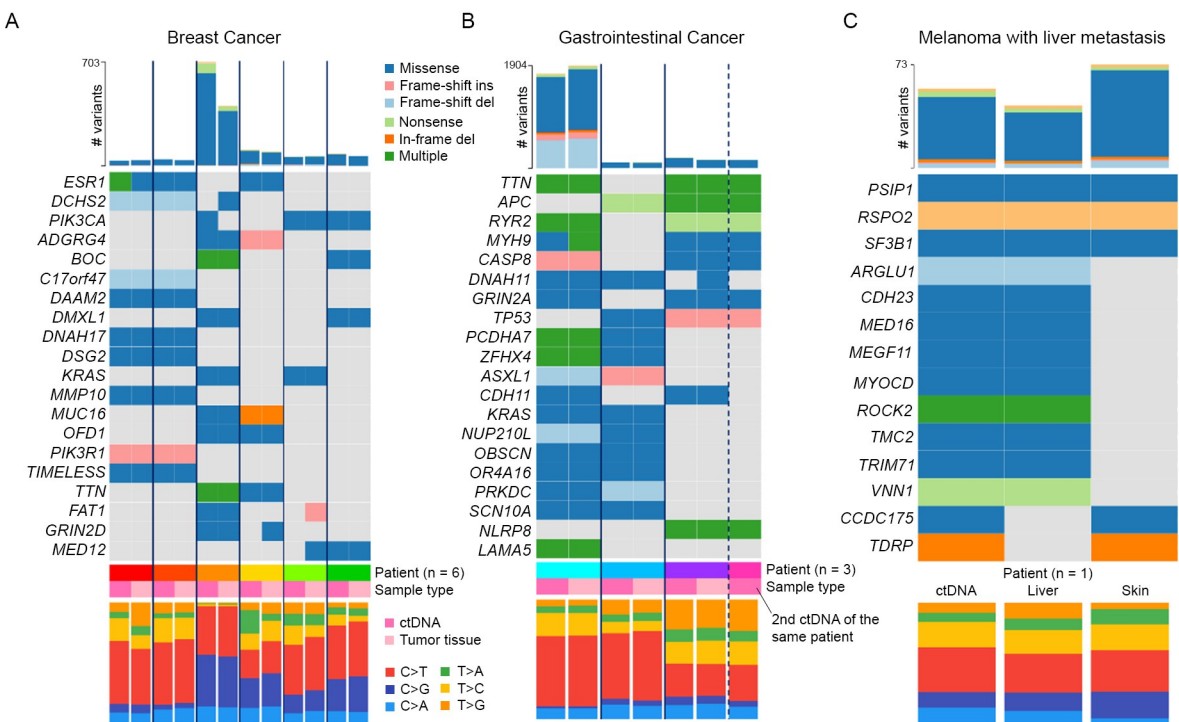

**Fig 4. ctDNA WES identifies similar top cancer-related genes as tumor WES.** A) Heatmap showing the variants of top cancer-related genes (from Cancer Gene Census via COSMIC database) identified in six breast cancer patients. Genes were ordered by observed mutation frequencies (%, shown on the y-axis with colors representing mutation types). Data for paired ctDNA and tumor tissue from the same patient were placed next to each other, with solid vertical bars separating data from different patients. Tumor mutational burdens (TMB) are indicated at the top. Base substitution profiles are shown at the bottom with legend located on the right-hand side. B) Similar plot for three gastrointestinal cancer patients with paired ctDNA and tumor WES data. Dashed vertical line designate the patient (GC01) with multiple ctDNA samples (GC01.1 and GC01.2). C) Similar plot for the melanoma patient with two tumor sites.

FAT1 and MED12 variants in ctDNA of patient BC08 (green) and the gain of PIK3CA variant in ctDNA WES of patient BC02 (orange), do not appear to be correlated low VAFs (S4 Table in S1 File, VAFs 16–28%) but may be attributed to low ctDNA fractions (60% for BC08 and 39% for BC02). The detailed list of tier-1 variants found in breast cancer patients is provided in S8 Table in S1 File.

For gastrointestinal cancer patients, 8 tier-1 genes, namely APC, CASP8, GRIN2A, MYH9, TP53, ASXL1, CDH11, and KRAS, were consistently identified by both ctDNA WES and tumor WES as the most frequently mutated genes (Fig 4B). Additional variants, including MUC5B, DNAH11, ANK2, and LAMA5, which were not part of tier-1 genes but has been reported in other gastrointestinal patients (according to the Cancer Browser [25]) were also consistently detected. There are only two discrepancies among these genes. The first was a minor variant of MYH9 with 9% VAF in the tumor that was missed by ctDNA WES (S5 Table in S1 File, c.1565del). It should be noted that the major MYH9 variant, c.1175C>T, was detected with >40% VAFs in both samples. The second discrepancy is a c.10654G>T substitution in DNAH11 with 17% VAF in the tumor tissue that was missed by both rounds of ctDNA WES. The detailed list of tier-1 variants found in gastrointestinal cancer patients is provided in S9 Table in S1 File.

For the melanoma patient with two tumor sites, three tier-1 genes, namely PSIP1, RSPO2, and SF3B1, were identified in both tumor tissues and ctDNA (Fig 4C). Three additional variants of tier-1 genes (BRCA2, MAF, and HOXD11) were identified exclusively in one sample.

The detailed list of tier-1 variants found in the melanoma patient is provided in S10 Table in S1 File.

## Discussion

In this study, we assessed the impact of ctDNA fraction on the quality of whole exome sequencing (WES) of ctDNA samples, analyzed the degree of concordance between WES of ctDNA and tumor samples, and showed the potential drift in mutational profiles over time when ctDNA and tumor tissues were collected several months apart. The results suggest that an appropriate cutoff for ctDNA fraction to ensure high quality WES data may be around 16–30% (Fig 1 and S2 Table in S1 File), since higher ctDNA fraction does not significantly improve the concordance with tumor WES data. This is in contrast with previous reports [17, 30] that recommended a rather low cutoff of 5–10% ctDNA fraction for performing WES. However, since the majority of ctDNA samples that were subjected to WES in these studies exhibited much higher than 10% ctDNA fractions (median ctDNA fraction of 30.8%, 35 out of 41 samples with greater than 20% ctDNA fractions) [17], the evidence for pinpointing a ctDNA fraction threshold that would ensure good WES quality is still lacking.

Similar to our finding that samples with ctDNA fractions below 16.4% (7.20–16.42%) yielded poor WES quality, low concordances between ctDNA and tumor tissue had been reported in samples with the same ctDNA fraction ranges. In Sabatier, R. *et al.* [31], the average ctDNA fraction was 12.7%, and all 11 samples with Pearson's correlation coefficients between ctDNA's and tumor tissue's copy number alteration profiles of 0.3 or lower were those with ctDNA fraction of less than 15%. Other studies have speculated that low ctDNA fractions were behind poor data quality [32, 33], but the exact ctDNA fractions were not measured or reported. Another consideration that affects a good cutoff for ctDNA fraction is the objective of ctDNA analysis. For tumor DNA detection, as low as 3% ctDNA fraction was reportedly enough [17]. For mutational load calculation [33] and copy number alteration analysis [31], data quality issues began to emerge in samples with low ctDNA fractions. For WES, ctDNA fraction may be an explanation for the wide variation in reported levels of concordance between ctDNA's and tumor tissue's mutational profiles [34, 35].

The observation that 4 out of 9 breast cancer samples exhibited low ctDNA fractions while none of the four gastrointestinal cancer samples did also suggest that the baseline ctDNA fraction, and consequently WES quality, can differ considerably across cancer tissues [35]. Although our cohort size is small and there are other factors, such as prior treatment history and tumor mutational burden [34], that can explain the low ctDNA WES quality, our findings strongly indicate that the cutoff for ctDNA fraction needs to be carefully re-examined, possibly separately for each cancer type.

Although WES of ctDNA and tumor may identify similar sets of variants with highly correlated variant allele frequencies (VAF) with Pearson's correlation reaching 0.92 in some patients (Fig 3 and S3 Table in S1 File), the actual VAFs can differ quite a lot (e.g., patients BC08 in Fig 3A and GC03 in Fig 3D). This does not necessarily indicate poor quality of ctDNA WES because tumor DNA in ctDNA may be shed from a different tumor region with a different mutational profile than the biopsied site due to high intra-tumor heterogeneity [36–39]. Even when the WES of tumor tissue perfectly captures the mutational profile of the whole tumor, VAF in ctDNA sample can still be skewed if some cancer cell subpopulations shed more DNA into the bloodstream than the rest. Indeed, the lack of strong correlation between high ctDNA fraction and high concordance in VAFs or high Jaccard Index of shared variants between ctDNA and tumor samples suggested that the quality of ctDNA sample is not the issue here.

As the time interval between ctDNA and tumor tissue sample collections increases, there is a clear, but non-significant decrease in concordances between ctDNA and tumor WES both in terms of allele frequencies and the number of shared variants. This patterns agreed well with previously reports that short time intervals (less than 3–6 months) did not impact the concordance [34, 40], while longer time intervals (more than 6 months) may have some effects [41]. This drop in concordances likely reflects the drift in mutational profiles as the tumor progresses, which reiterates the need for an inexpensive, non-invasive technique that can be repeatedly performed to monitor the molecular status of the tumor over time. It is interesting to note that in our cohort the temporal drift in mutational profile can explain the degree of concordance between ctDNA and tumor tissue slightly better than the tumor fraction of ctDNA does.

WES analysis of ctDNA can capture mutational profiles from multiple tumor sites within the same patient [33, 35]. Here, all 22 shared variants between the two tumor sites in the melanoma patient (MN01) were identified in ctDNA, together with 12 additional unique variants from either site. The higher concordance between ctDNA and the metastatic liver site compared to the primary skin site may be due to two reasons. First, more DNA molecules may be shed from liver metastatic cells into the bloodstream [35]. Second, the higher VAFs and lower number of unique variants from metastatic site indicate that the metastatic tumor is less heterogeneous than the primary tumor. This factor may increase the relative concentration of metastatic variants in ctDNA sample.

Lastly, WES of ctDNA can consistently capture cancer-related genes of interest [27] that were present in tumor tissues (Fig 4), including ESR1, KRAS, PIK3CA, PIK3R1, FAT1, MED12, and MUC16 for breast cancer, APC, CASP8, TP53, KRAS, CDH11, GRIN2A, ASXL1 and MYH9 for gastrointestinal cancer, and PSIP1, RSPO2, and SF3B1 for melanoma/liver cancer. WES of ctDNA can also identify additional, potentially actionable, genes of interest not found in tumor WES results [33, 34], but further validations would be required. Indeed, several prior studies have shown that WES of ctDNA identified many unique variants not found in WES of tumor tissues [33, 35] and that combining WES of multiple sample sources yield a more complete profile of actionable mutations [30]. Diefenbach, R.J. *et al.* [35] reported 22.7–77.6% overlap in called mutations between ctDNA and tumor tissue while Koeppel, F. *et al.* [33] reported that as many as 47% of ctDNA variants cannot be detected in WES of tumor tissues. Hence, these findings affirm that WES of ctDNA is reliable and valuable for probing the genetic profiles of tumor.

WES of ctDNA can also identify similar subclones detected in WES of tumor tissue, as well as additional subclones, in many cases (S4 Fig). Furthermore, variants that constitute the same subclone tend to be detected at similar VAFs in both WES results. However, the concordance between the sets of variants that constitute the same subclone in ctDNA and tumor tissue are generally quite low (Jaccard Index lower than 0.2), indicating that different subclonal events were detected in each sample. Our results are in good agreement with prior studies which established that while clonal variants can be reliably characterized in both WES of tumor tissue and ctDNA [17, 30] (88–99% concordance on average), different subclonal events were detected (45% concordance on average). Therefore, although more subclones can be identified via WES of ctDNA, the fact that they were associated with different sets of variants from those found in tumor biopsies means that they might lead to a different clinical interpretation and must be carefully validated.

There are three main limitations of this study. First, the sample size and cancer variety are not high enough to establish statistical significance on each of the factors that contributed to the quality of ctDNA analysis. Second, the patient cohort was not systematically structured to allow independent investigations into each factor that contributed to the quality of ctDNA

analysis while controlling for the other factors. A future experiment that fixes the length of time interval between sample collection would provide a clearer assessment of the impact of ctDNA fraction. Likewise, a future experiment that collected ctDNA samples from multiple time points for the same patient would provide a better evaluation of the drift in mutational profile over time. Purposely mixing plasma samples with known ratios of PBMCs, experimentally or *in silico* [17], would also enable a well-controlled evaluation of the impact of ctDNA fraction on the quality of WES results. Third, the lengths of time interval between the first tumor tissue biopsy and ctDNA collection were rather long in several cases. These limitations stemmed from logistical issues and patient health issues that prevented us from obtaining samples in a timely manner.

Overall, our study presents another real-world application of whole exome sequencing of ctDNA samples on three patient groups that reiterates the need for better understanding of the impact of cell-free DNA sample (cfDNA) characteristics on the quality of subsequent WES analyses. The ability to ensure high quality of tumor variant characterizations is important, especially for WES, as it incurs cost on the patients. Our findings indicate that a recommended cutoff for ctDNA fraction for WES of 5–10% [17, 30] may be too low [31] and that the appropriate cutoff may be cancer tissue-specific (Fig 2A), possibly due to the different shedding of tumor cells from various tissues into the bloodstream [35]. Although the high concordance between variants of top cancer-related genes (Fig 4) in WES of ctDNA and tumor tissue may suggest that WES of ctDNA alone is enough for clinical applications, the large numbers of unique variants in WES of both samples (S3 Table in S1 File) indicate that performing WES on both samples can uncover considerably more information.

## Supporting information

**S1 Fig. Summary of sample collection time points.** Time intervals between tumor tissue WES and ctDNA WES are indicated. Low quality ctDNA WES from samples with low ctDNA fractions were not considered in subsequent analyses.
(TIF)

**S2 Fig. Copy number ratios of 4 patients with ctDNA fractions lower than 30%.** These patients were excluded from subsequent concordance analyses.
(TIF)

**S3 Fig. Variant allele frequencies of non-synonymous variants identified by only WES of ctDNA, only WES of tumor tissue, or both.** A) VAFs calculated from WES of tumor tissue. B) VAFs calculated from WES of ctDNA. Data from breast cancer (BC) and gastrointestinal cancer (GC) were shown separately as the two cancer types exhibit different VAF levels. All comparisons of VAFs between shared (concordant) and tumor-only or ctDNA-only (discordant) are statistically significant (Mann-Whitney U test p-values $< 2.14e-5$).
(TIF)

**S4 Fig. Subclone VAF profiles in WES of tumor tissue and ctDNA of selected breast cancer and gastrointestinal cancer samples.** A) Scatter plot showing sequencing depths and VAFs for variants identified in BC01 patient's tumor sample. Coloring indicates different subclones. B) Scatter plot showing sequencing depths and VAFs for variants identified in BC01 patient's ctDNA sample. C) Heatmap showing the relative Jaccard Index between subclones identified in tumor and ctDNA samples from BC01 patient. D-F) Similar plots for BC03 patient. G-I) Similar plots for GC01 patient. J-L) Similar plots for GC03 patient.
(TIF)

**S1 File.**
(XLSX)

## Author Contributions

**Conceptualization:** Trairak Pisitkun, Piriya Wongkongkathep.

**Data curation:** Prangwalai Chanchaem, Suwanan Mankhong, Sikrit Denariyakoon, Valla Fongchaiya, Areeya Arayataweegool, Pattama Angspatt, Ploytuangporn Wongchanapai, Verayuth Prapanpoj, Kris Chatamra.

**Formal analysis:** Julanee Leenanitikul, Sira Sriswasdi.

**Investigation:** Julanee Leenanitikul.

**Resources:** Prangwalai Chanchaem, Suwanan Mankhong, Sikrit Denariyakoon, Valla Fongchaiya, Areeya Arayataweegool, Pattama Angspatt, Ploytuangporn Wongchanapai, Verayuth Prapanpoj, Kris Chatamra.

**Supervision:** Sira Sriswasdi, Piriya Wongkongkathep.

**Visualization:** Julanee Leenanitikul.

**Writing – original draft:** Julanee Leenanitikul.

**Writing – review & editing:** Trairak Pisitkun, Sira Sriswasdi, Piriya Wongkongkathep.

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
