## [Decision Letter · Decision Letter 0]

31 Jul 2023

PONE-D-23-16936Concordance between whole exome sequencings of circulating tumor DNA and tumor tissuePLOS ONE

Dear Dr. Sriswasdi,

Thank you for submitting your manuscript to PLOS ONE. After careful consideration, we feel that it has merit but does not fully meet PLOS ONE’s publication criteria as it currently stands. Therefore, we invite you to submit a revised version of the manuscript that addresses the points raised during the review process.

We look forward to receiving your revised manuscript.

Kind regards,

Alvaro Galli

Academic Editor

PLOS ONE

“This research was partially supported by the Thailand Science Research and Innovation Fund, Chulalongkorn University (HEA663000041 to S.S.), the grants for development of new faculty staff, Ratchadaphiseksomphot Fund, Chulalongkorn University (to P.W.), and Ratchadaphiseksomphot Fund, Faculty of Medicine, Chulalongkorn University, grant number RA65/029 (to P.W., S.S., and T.P.)”

“This research was partially supported by the Thailand Science Research and Innovation Fund, Chulalongkorn University (HEA663000041 to S.S.), the grants for development of new faculty staff, Ratchadaphiseksomphot Fund, Chulalongkorn University (to P.W.), and Ratchadaphiseksomphot Fund, Faculty of Medicine, Chulalongkorn University, grant number RA65/029 (to P.W., S.S., and T.P.). The funders had no role in study design, data collection and analysis, decision to publish, or preparation of the manuscript.”

3. We noted in your submission details that a portion of your manuscript may have been presented or published elsewhere. [Yes, a shorter version of the manuscript (contain results for a subset of the patients) is under consideration for a proceeding at a local conference in Thailand. The conference is organized in Thai language and its proceeding is not indexed by any international database. The submission to the conference was made solely to fulfill the master’s degree graduation requirement of the first author, Ms. Leenanitikul. We have uploaded the related manuscript along with this submission for full transparency.] Please clarify whether this conference proceeding was peer-reviewed and formally published. If this work was previously peer-reviewed and published, in the cover letter please provide the reason that this work does not constitute dual publication and should be included in the current manuscript.

5. Please amend either the title on the online submission form (via Edit Submission) or the title in the manuscript so that they are identical.

Reviewers' comments:

Reviewer's Responses to Questions

**Comments to the Author**

1. Is the manuscript technically sound, and do the data support the conclusions?

Reviewer #1: Yes

Reviewer #2: Yes

Reviewer #3: Yes

2. Has the statistical analysis been performed appropriately and rigorously? 

Reviewer #1: Yes

Reviewer #2: Yes

Reviewer #3: Yes

3. Have the authors made all data underlying the findings in their manuscript fully available?

Reviewer #1: Yes

Reviewer #2: Yes

Reviewer #3: Yes

4. Is the manuscript presented in an intelligible fashion and written in standard English?

Reviewer #1: Yes

Reviewer #2: Yes

Reviewer #3: Yes

5. Review Comments to the Author

Reviewer #1: Authors demonstrate that Whole-Exome Sequencing of circulating tumor DNA detected cancer-related genes. The findings probe that Whole-Exome Sequencing of circulating tumor DNA is reliable and valuable.

Overall, the manuscript is well written.

Reviewer #2: The science in this manuscript and its interpretation are fine as far as they go, but there are some limitations to the article, which mean that I cannot recommend publication without revision. To the credit of the authors, they do mention some of these limitations in the discussion, although the discussion could go into more depth and reference other papers using ctDNA which do deal with these issues.

Firstly, the study uses a relatively small number of patients with one of four types of cancer. I am very aware that it may not always be possible to obtain as many samples from donors as desired, but I feel that the manuscript would work much better if it were more tightly focussed on samples taken from one type of solid cancer (such as breast cancer or GI cancer) rather than the four diseases they chose. As it stands, using different cancers mean that the ctDNA tumour fraction threshold determined for accurate concordance between variants here may be on the high side as it must apply across all four disease types, when the threshold may be different for each one. Also, and this is mentioned within the discussion, a more consistent approach to taking the follow-up blood samples (for cfDNA analysis) at a fixed window after the tissue samples would massively help with the interpretation of how concordance changes over time in the ctDNA.

Secondly, the authors do mention that the lower bound cut-off of 16.7% tumour fraction within ctDNA could be more robustly determined by analysing a dilution series. I am not sure if there are enough samples to do this, but an additional experiment in which fixed, known, ratios of plasma (for the cfDNA fraction) and PBMCs (for the germline fraction) would enable the ichorCNA (for tumour fraction calculations) and variant calling pipelines to be thoroughly separately tested and validated.

Finally, the cut-off for sequencing depth and number of reads supporting the variant appear quite high – Mouliere et al (2018, Sci Trans Med. 2018 Nov 7;10(446), doi:10.1126/scitranslmed.aat4921) use a minimum depth for variant calling of three reads (compared to eight in this study), with one read supporting the variant in each direction (compared to a total of four reads supporting the variant in this study). I am not sure whether the choice of such a high cut-off is justified within the manuscript and would like to know how these values were decided upon. It is possible that the read depth thresholds being so high mean that the tumour fraction cut-off must be as high as 16.7% to capture a significant fraction of the concordant variants between the ctDNA and tumour samples. A figure showing how the concordance varies with the minimum read cut-offs in the supplementary data would help to justify this.

Specific points:

- Title: I feel that the “s” at the end of “whole exome sequencings” is not required

- Supplementary Table 1 would be better supplemented with a graphic showing the timeline of the individual patients and samples. See for instance, figure 1B in Sivapalan et al (2022), BMC Cancer, 22:369, doi: 10.1186/s12885-022-09387-6.

Reviewer #3: Review of PONE-D-23-16936: Concordance between whole exome sequencings of circulating tumor DNA and tumor tissue by Leenanitikul, J et al.

In this manuscript the authors have conducted a validation study of whole-exome sequencing (WES) on paired ctDNA and tumor biopsy in 15 cancer patients to assess the extent of concordance between mutational profiles derived from the two source materials. The primary conclusion of the study is to counter previously a published report that that have stated a much lower cutoff for ctDNA fraction at 3% and recommend instead a range of 16-30% cutoff.

Figure 1 and Table 1 are clear and interpretable. Figure 2 needs to be larger and have legends for the different color-coded points in the scatter-graph. On line 206 the authors state that “There is a clear positive correlation between ctDNA fraction and the number of identified variants (Figure 2A, Pearson’s correlation = 0.4657, p-value = 0.069).” However, the correlation coefficient is low, and the p-value does not reach significance, this must be commented upon.

While the references list citations 34-36, these are not referred to in the text of the manuscript. Also, the Discussion should include a comparison of the authors findings in the context of other similar studies that have compared WES in ctDNA vs. tumor biopsy, such as references 34-36, which though listed in the references, are not discussed, or cited in the main text. The authors may also consider referring to Imperial et al, 2019 (PMID: 31546879), Sabatier et al 2022 (PMID: 35965534) and Manier et al 2018 (PMID: 29703982). The cut-off ranges for ctDNA fractions across these publications would also serve to bolster the authors conclusions.

Figure 4 is well presented however, again, the legends of the figure are impossibly small which makes ease of interpretation difficult. The authors should increase the size. Plus, simple titles such as “Breast cancer, n=6”; “Gastric cancer, n=3” and “Melanoma n=1” would go a long way for ease of interpretation.

The authors should discuss the clonality or sub-clonality of the mutations observed and whether it is possible capture sub-clonal variants in both methodologies. Lastly, the authors may consider discussing the clinical applicability of their findings, the clinical scenarios in which ctWES should be considered over and above that of tumor biopsy WES in the context of their findings.

6. PLOS authors have the option to publish the peer review history of their article (what does this mean?). If published, this will include your full peer review and any attached files.

Reviewer #1: No

Reviewer #2: No

Reviewer #3: **Yes: **D Chakravarty

---

## [Author Response · Author response to Decision Letter 0]

21 Sep 2023

Response: We have reformatted the manuscript according to the guidelines.

“This research was partially supported by the Thailand Science Research and Innovation Fund, Chulalongkorn University (HEA663000041 to S.S.), the grants for development of new faculty staff, Ratchadaphiseksomphot Fund, Chulalongkorn University (to P.W.), and Ratchadaphiseksomphot Fund, Faculty of Medicine, Chulalongkorn University, grant number RA65/029 (to P.W., S.S., and T.P.)”

“This research was partially supported by the Thailand Science Research and Innovation Fund, Chulalongkorn University (HEA663000041 to S.S.), the grants for development of new faculty staff, Ratchadaphiseksomphot Fund, Chulalongkorn University (to P.W.), and Ratchadaphiseksomphot Fund, Faculty of Medicine, Chulalongkorn University, grant number RA65/029 (to P.W., S.S., and T.P.). The funders had no role in study design, data collection and analysis, decision to publish, or preparation of the manuscript.”

Response: We have addressed this issue in the cover letter and removed all funding-related statements from the Acknowledgement section in the manuscript.

3. We noted in your submission details that a portion of your manuscript may have been presented or published elsewhere. [Yes, a shorter version of the manuscript (contain results for a subset of the patients) is under consideration for a proceeding at a local conference in Thailand. The conference is organized in Thai language and its proceeding is not indexed by any international database. The submission to the conference was made solely to fulfill the master’s degree graduation requirement of the first author, Ms. Leenanitikul. We have uploaded the related manuscript along with this submission for full transparency.] Please clarify whether this conference proceeding was peer-reviewed and formally published. If this work was previously peer-reviewed and published, in the cover letter please provide the reason that this work does not constitute dual publication and should be included in the current manuscript.

Response: We have clarified in the cover letter that the conference offered an option to revise the manuscript and publish it in their scientific journal. We did not choose this option and only included our shorter manuscript as part of the proceedings, which sufficed for the graduation requirement of Ms. Leenanitikul. The original statement supporting our explanation on the conference website and the corresponding Thai-to-English translation are provided in the cover letter.

Response: We have provided a full Data Availability statement to show that the minimal data set required to reanalyze and reproduce the findings is available in Supplementary Tables. The specific information contained within each table was explained in detail. Raw sequencing data were not released because patient’s exome sequencing data are considered the patient’s private information. A statement that a request for raw data may be made to the corresponding author was also added [Lines 570-578].

5. Please amend either the title on the online submission form (via Edit Submission) or the title in the manuscript so that they are identical.

Response: We have checked this issue.

 

Response to review comments

Overall Response: We would like to thank the reviewers for their suggestions which help us improve the quality of the manuscript. Please find our point-by-point responses below along with the corresponding line numbers in the revised manuscript.

Reviewer #1: Authors demonstrate that Whole-Exome Sequencing of circulating tumor DNA detected cancer-related genes. The findings probe that Whole-Exome Sequencing of circulating tumor DNA is reliable and valuable.

Overall, the manuscript is well written.

Response: Thank you very much. We have further improved the manuscript based on suggestions from the other reviewers as well.

Reviewer #2: The science in this manuscript and its interpretation are fine as far as they go, but there are some limitations to the article, which means that I cannot recommend publication without revision. To the credit of the authors, they do mention some of these limitations in the discussion, although the discussion could go into more depth and reference other papers using ctDNA which do deal with these issues.

Response: Thank you for your understanding of our limitations. We highly appreciated the suggestion of related works which add to the comparison and discussion of our results.

Firstly, the study uses a relatively small number of patients with one of four types of cancer. I am very aware that it may not always be possible to obtain as many samples from donors as desired, but I feel that the manuscript would work much better if it were more tightly focused on samples taken from one type of solid cancer (such as breast cancer or GI cancer) rather than the four diseases they chose. As it stands, using different cancers mean that the ctDNA tumour fraction threshold determined for accurate concordance between variants here may be on the high side as it must apply across all four disease types, when the threshold may be different for each one. 

Response: Thank you for the suggestion on the cancer-specificity viewpoint. We have added the observation that almost all samples with low ctDNA fractions and poor WES quality (4 out of 5) are breast cancers (4 out of 9 breast cancer samples have low ctDNA fractions) while none of the four gastrointestinal samples exhibits low ctDNA fraction. Although the sample size is small, this suggests that cancer type may have an effect and should be considered as suggested by the reviewer. [Lines 190-191, 348-354]

Also, and this is mentioned within the discussion, a more consistent approach to taking the follow-up blood samples (for cfDNA analysis) at a fixed window after the tissue samples would massively help with the interpretation of how concordance changes over time in the ctDNA.

Response: We were hindered by logistical and patient health issues that made us unable to control the sample collection timing. However, we tried to make the best use of this dataset to better understand the characteristics of WES analysis of ctDNA. A discussion about the impact of time interval between samples collection has been added. [Lines 368-370]

Secondly, the authors do mention that the lower bound cut-off of 16.7% tumour fraction within ctDNA could be more robustly determined by analyzing a dilution series. I am not sure if there are enough samples to do this, but an additional experiment in which fixed, known, ratios of plasma (for the cfDNA fraction) and PBMCs (for the germline fraction) would enable the ichorCNA (for tumour fraction calculations) and variant calling pipelines to be thoroughly separately tested and validated.

Response: Thank you for your suggestion. We have revised the statement to mention the mixing of plasma samples and PBMC instead. We also cited the ichorCNA paper which did this approach with in silico mixing of reads from plasma and PBMC samples. [Lines 417-421]

Finally, the cut-off for sequencing depth and number of reads supporting the variant appear quite high – Mouliere et al (2018, Sci Trans Med. 2018 Nov 7;10(446), doi:10.1126/scitranslmed.aat4921) use a minimum depth for variant calling of three reads (compared to eight in this study), with one read supporting the variant in each direction (compared to a total of four reads supporting the variant in this study). I am not sure whether the choice of such a high cut-off is justified within the manuscript and would like to know how these values were decided upon. It is possible that the read depth thresholds being so high mean that the tumour fraction cut-off must be as high as 16.7% to capture a significant fraction of the concordant variants between the ctDNA and tumour samples. A figure showing how the concordance varies with the minimum read cut-offs in the supplementary data would help to justify this.

Response: Thank you for raising this point. We tried relaxing the variant filtering threshold down to the lowest values of variant read(s) ≥ 1 (from 4) and depth ≥ 2 (from 8), but only 1-2 additional variants were called for these samples. Hence, the low numbers of called variants in these samples cannot be explained by the quality filtering step. [Lines 196-197]

We also checked the supplementary methods of the suggested paper and other references more carefully and found that the suggested paper also required at least 4 reads supporting the variant, as quoted below.

“Mutation calling. […]. After MuTect2, we applied filtering parameters so

that a mutation was called if no mutant reads for an allele were observed in germline

DNA at a locus that was covered at least 10x, and if at least 4 reads supporting the

mutant were found in the plasma data with at least 1 read on each strand (forward

and reverse). […].”

 But Adalsteinsson, V.A. et al. PMID: 29109393 did use a minimum number of reads supporting the variant of 3.

Specific points:

- Title: I feel that the “s” at the end of “whole exome sequencings” is not required

Response: We have removed this “s”.

- Supplementary Table 1 would be better supplemented with a graphic showing the timeline of the individual patients and samples. See for instance, figure 1B in Sivapalan et al (2022), BMC Cancer, 22:369, doi: 10.1186/s12885-022-09387-6.

Response: We have added a new Supplementary Figure 1 (S1 Fig) that clearly shows the timeline of sample collections and the time interval between tumor tissue WES and ctDNA WES of the same patient.

Reviewer #3: In this manuscript the authors have conducted a validation study of whole-exome sequencing (WES) on paired ctDNA and tumor biopsy in 15 cancer patients to assess the extent of concordance between mutational profiles derived from the two source materials. The primary conclusion of the study is to counter previously a published report that that have stated a much lower cutoff for ctDNA fraction at 3% and recommend instead a range of 16-30% cutoff.

Figure 1 and Table 1 are clear and interpretable. Figure 2 needs to be larger and have legends for the different color-coded points in the scatter-graph. 

Response: Figure 2 has been revised to clearly distinguish samples from different cancer types.

On line 206 the authors state that “There is a clear positive correlation between ctDNA fraction and the number of identified variants (Figure 2A, Pearson’s correlation = 0.4657, p-value = 0.069).” However, the correlation coefficient is low, and the p-value does not reach significance, this must be commented upon.

Response: We added an explanation that the correlation coefficient did not reach significance likely due to the high variation and low number of identified variants among samples with ctDNA above 50% (Fig 2A). [Lines 185-188]

While the references list citations 34-36, these are not referred to in the text of the manuscript.

Response: We are sorry for the oversight. The discussions involving these manuscripts were somehow deleted. We have expanded the discussion to incorporate these references and additional references suggested below. [Lines 331-347, 368-370, 392-410]

Also, the Discussion should include a comparison of the authors findings in the context of other similar studies that have compared WES in ctDNA vs. tumor biopsy, such as references 34-36, which though listed in the references, are not discussed, or cited in the main text. The authors may also consider referring to Imperial et al, 2019 (PMID: 31546879), Sabatier et al 2022 (PMID: 35965534) and Manier et al 2018 (PMID: 29703982). The cut-off ranges for ctDNA fractions across these publications would also serve to bolster the authors conclusions.

Response: Thank you very much for the suggested articles. We have incorporated the findings regarding ctDNA fraction and the level of concordance between results from ctDNA and tumor tissue. We feel that the discussion became much richer. [Lines 331-347, 368-370, 392-410]

Figure 4 is well presented however, again, the legends of the figure are impossibly small which makes ease of interpretation difficult. The authors should increase the size. Plus, simple titles such as “Breast cancer, n=6”; “Gastric cancer, n=3” and “Melanoma n=1” would go a long way for ease of interpretation.

Response: We have revised Fig. 4 and increased the size of all legends.

The authors should discuss the clonality or sub-clonality of the mutations observed and whether it is possible capture sub-clonal variants in both methodologies. 

Response: We have performed subclone analysis using MAGOS tool (Ahmadinejad, N. et al. Mol Biol Evol 39(7): msac136, 2022) and added the results for representative breast cancer and gastrointestinal samples in Supplementary Figure 4. The results showed that there is some correspondence between subclones identified in tumor tissue and those found in ctDNA. However, the concordance in variants is low, which is similar to prior studies which reported only 45% concordance in subclonal variants between tumor tissue and ctDNA. These points have been added to the discussion. [Lines 399-410]

Lastly, the authors may consider discussing the clinical applicability of their findings, the clinical scenarios in which ctWES should be considered over and above that of tumor biopsy WES in the context of their findings.

Response: We have added this discussion point to the revision. [Lines 342-347, 426-436]

---

## [Decision Letter · Decision Letter 1]

2 Oct 2023

Concordance between whole exome sequencing of circulating tumor DNA and tumor tissue

PONE-D-23-16936R1

Dear Dr. Sriswasdi,

We’re pleased to inform you that your manuscript has been judged scientifically suitable for publication and will be formally accepted for publication once it meets all outstanding technical requirements.

Kind regards,

Alvaro Galli

Academic Editor

PLOS ONE

Additional Editor Comments (optional):

Reviewers' comments:

Reviewer's Responses to Questions

**Comments to the Author**

1. If the authors have adequately addressed your comments raised in a previous round of review and you feel that this manuscript is now acceptable for publication, you may indicate that here to bypass the “Comments to the Author” section, enter your conflict of interest statement in the “Confidential to Editor” section, and submit your "Accept" recommendation.

Reviewer #1: (No Response)

Reviewer #2: All comments have been addressed

2. Is the manuscript technically sound, and do the data support the conclusions?

Reviewer #1: Yes

Reviewer #2: Yes

3. Has the statistical analysis been performed appropriately and rigorously? 

Reviewer #1: Yes

Reviewer #2: Yes

4. Have the authors made all data underlying the findings in their manuscript fully available?

Reviewer #1: Yes

Reviewer #2: Yes

5. Is the manuscript presented in an intelligible fashion and written in standard English?

Reviewer #1: Yes

Reviewer #2: Yes

6. Review Comments to the Author

Reviewer #1: (No Response)

Reviewer #2: All my comments have been addressed, as have those of the other reviewers. I am happy for the manuscript to be published in its current form.

7. PLOS authors have the option to publish the peer review history of their article (what does this mean?). If published, this will include your full peer review and any attached files.

Reviewer #1: No

Reviewer #2: No

---

## [Editor Report · Acceptance letter]

17 Oct 2023

PONE-D-23-16936R1 

Concordance between whole exome sequencing of circulating tumor DNA and tumor tissue 

Dear Dr. Sriswasdi:

I'm pleased to inform you that your manuscript has been deemed suitable for publication in PLOS ONE. Congratulations! Your manuscript is now with our production department. 

Kind regards, 

on behalf of

Dr. Alvaro Galli 

Academic Editor

PLOS ONE